# Synthesis of Novel Antimicrobial CHX-CaCl_2_ Coatings on Maxillofacial Fixatures for Infection Prevention

**DOI:** 10.3390/ijms24129801

**Published:** 2023-06-06

**Authors:** Hawraa F. Alostath, Domniki Chatzopoulou, Simon Holmes, David Gould, Gleb Sukhorukov, Michael J. Cattell

**Affiliations:** 1Centre for Oral Bioengineering, Bart’s and the London, School of Medicine and Dentistry, Queen Mary University of London, London E1 2AD, UK; h.alostath@qmul.ac.uk (H.F.A.); d.chatzopoulou@qmul.ac.uk (D.C.); 2Barts Health NHS Trust, Department of Oral and Maxillofacial Surgery, Queen Mary University of London, London E1 1FR, UK; s.holmes@qmul.ac.uk; 3Centre for Biochemical Pharmacology, William Harvey Research Institute, Queen Mary University of London, London EC1M 6BQ, UK; d.j.gould@qmul.ac.uk; 4Institute of Bioengineering, School of Engineering and Materials Science, Queen Mary University of London, London E1 4NS, UK; g.sukhorukov@qmul.ac.uk

**Keywords:** chlorhexidine, maxillofacial, antimicrobial, drug delivery, crystallisation, drug synthesis

## Abstract

Maxillofacial surgery placement of fixatures (Leonard Buttons, LB) at close proximity to surgical incisions provides a potential reservoir as a secondary local factor to advanced periodontal disease, with bacterial formation around failed fixatures implicating plaque. To address infection rates, we aimed to surface coat LB and Titanium (Ti) discs using a novel form of chlorhexidine (CHX), CHX-CaCl_2_ and 0.2% CHX digluconate mouthwash as a comparison. CHX-CaCl_2_ coated, double-coated and mouthwash coated LB and Ti discs were transferred to 1 mL artificial saliva (AS) at specified time points, and UV-Visible spectroscopy (254 nm) was used to measure CHX release. The zone of inhibition (ZOI) was measured using collected aliquots against bacterial strains. Specimens were characterized using Energy Dispersive X-ray Spectroscopy (EDS), X-ray Diffraction (XRD) and Scanning Electron Microscopy (SEM). SEM displayed copious dendritic crystals on LB/ Ti disc surfaces. Drug release from double-coated CHX-CaCl_2_ was 14 days (Ti discs) and 6 days (LB) above MIC, compared to the comparison group (20 min). The ZOI for the CHX-CaCl_2_ coated groups was significantly different within groups (*p* < 0.05). CHX-CaCl_2_ surface crystallization is a new drug technology for controlled and sustained CHX release; its antibacterial effectiveness makes this drug an ideal adjunct following clinical and surgical procedures to maintain oral hygiene and prevent surgical site infections.

## 1. Introduction

Maxillofacial trauma has a common and acute presentation in accident and emergency departments, with the incidence increased within the UK from 4% (1998) to 10% (2020), according to a BOAMS survey [1,2]. Mandibular fractures are prone to infection, and strategies to reduce surgical site infection remain ineffective and are targeted at smoking cessation and alcohol withdrawal, with no proven efficacy [3,4]. Surgical site infection remains a considerable problem with increased cost and morbidity to the patient [3]. Recently, our group demonstrated that advanced periodontal disease was a highly significant cofactor in surgical site infection (mandibular fractures) with an odds ratio of 74. Maxillofacial plate removal is primarily due to infection in 46.7% of the cases [5,6] and results from biofilm development [7], which mandates further surgery.

Contemporaneous management of mandibular fracture involves maxillomandibular fixation (MMF). This commonly involves using wires and metal fixatures to localise the fractures following the application of surgical fixatures (e.g., arch bars and Leonard Buttons) [8,9,10,11], followed by intra-oral surgical access in the buccal sulcus and application of load-sharing osteosynthesis according to Champy principles. It is postulated that the dependent position of the surgical incision in the buccal sulcus makes the site vulnerable to biofilm-producing bacteria, and strategies to reduce this load may prove beneficial to fracture management.

Currently, a Chlorhexidine (CHX) digluconate mouthwash (0.12%, 0.2% and 1%) is used prior to and following craniomaxillofacial surgery and as a recommended longer-term postoperative mouthwash [12]. These mouth-rinse adjuncts were also recommended during the COVID-19 pandemic to minimize transmission upon treatment [13,14], and various detection methods were suggested [15]. Infection rates are, nevertheless, still high (4–10%) in MMF patients, and this needs improvement. A solution to this problem may be the use of a novel crystalline form of chlorhexidine (CHX-CaCl_2_) formulated by Luo et al. [16], which provides a safe and effective antimicrobial with substantivity. Novel CHX-CaCl_2_ particles also demonstrated a surface crystallisation mechanism when synthesised in conjunction with gold nanorods [17]. This technology may, therefore, have the potential to be applied to metallic MMF close to the fracture site.

The aim of this study was to synthesise a CHX-CaCl_2_particle antimicrobial coating onto MMF to improve oral hygiene and reduce infection rates in the maxillofacial traumatology that is lacking within research. A slow CHX release from the coated MMF may reduce infections and prevent plaque build-up, maintaining oral hygiene.

## 2. Results

### 2.1. Energy Dispersive X-ray Spectroscopy (EDS) Results

The EDS results for the titanium disc and the LB (Ti element) are shown in Figure 1a,b and the quantitative elemental composition (atomic %) is displayed in Table 1.

### 2.2. Scanning Electron Microscopy (SEM) Results

SEM photomicrographs presented in Figure 2 show the CHX-CaCl_2_ crystals on the single coated Ti disc (a–c), double-coated Ti disc (d–f), and lapped Ti disc (g–i). There was a dense dispersal of CHX-CaCl_2_ crystals (Figure 2a) binding dendritically to the Ti disc surfaces and linking to other crystals (Figure 2b,c). The double coating of Ti discs displayed a fine dispersal of CHX-CaCl_2_ crystals on the surface (Figure 2d), with partially formed crystallites growing on a bed of crystallites (Figure 2e). The CHX-CaCl_2_ crystal primary growth appeared to be from a central nucleus outwards in a wheat-sheaf appearance (Figure 2f). The lapped Ti disc displayed a dispersed coating (Figure 2g) with a bi-modal distribution of CHX-CaCl_2_ crystals (Figure 2h), where smaller crystal formation was associated with flaws on the lapped surface (Figure 2h,i). There was no coating observed for the Corsodyl-coated Ti disc surface (Figure 2j,k), which displayed the Ti disc’s coarse surface and grain boundaries (Figure 2k,l).

Figure 3a shows the uncoated Leonard Button (LB) displaying a coarse and columnar surface (Figure 3b). Figure 3c displays signs of some agglomerated spherical CHX-CaCl_2_ crystals on the LB’s concave surface and a uniform crystal distribution on its convex surface (Figure 3d). The CHX-CaCl_2_ crystals are growing in association with the LB’s surface flaws (Figure 3e). Figure 3f illustrates the unique spherical morphology of the crystals, with dendrites binding uniformly to the LB’s coarse metal surface. The intimate contact of the CHX-CaCl_2_ crystals’ tendrils at the LB surface interface is evident in Figure 3f.

### 2.3. Light Profilometry Results

The mean surface roughness (Ra) is indicated in Table 2 for the lapped and unlapped Ti discs. Figure 4 illustrates the primary and grid mapping of (a) the Ti disc and (b) the lapped Ti disc specimen.

### 2.4. Ultraviolet-Visible Spectroscopy (UV-Vis) Results

The calibration concentrations against absorbance were highly correlated for the Chlorhexidine diacetate (CHXD) and CHX digluconate standards (r^2^ = 1 and 0.99). Figure 5a,b demonstrates the cumulative CHXD release for the Ti disc Grps. 1–4 and Leonard Button Grps. 5–7. A rapid release of the CHXD is illustrated across all groups within the first 10 min, then an increase in CHXD release was observed after 24 h for the single coated CHX-CaCl_2_ (Grps. 1–3 and 5–6). Figure 5 shows a sustained release of CHXD up to 14 days (>2.5 ppm) for the double-CHX-CaCl_2_-coated Ti discs (Grp. 2) and up to 6 days (>2.5 ppm) for single CHX-CaCl_2_-coated Ti discs (Grp. 1). Single CHX-CaCl_2_ coating of LB resulted in a sustained CHXD release of 3 days and increased to 6 days for double coating (>2.5 ppm). However, the commercial mouthwash (Grps. 3 and 7) coated Ti discs and LB indicated a CHX release of 20 min (>2.5 ppm).

### 2.5. X-ray Diffraction (XRD) Results

The CHX-CaCl_2_ crystals displayed unique peaks synonymous with the patented form [18]. The XRD data indicate the 2-theta peaks in Figure 6.

### 2.6. Zone of Inhibition (ZOI) Results

The CHXD standard concentrations (ppm) were correlated with the mean diameter (mm) of the zone of inhibition for S. mutans (r^2^ = 0.98) and *P. gingivalis* (r^2^ = 0.97). ZOI results are shown in Figure 7a,b for Ti discs and Figure 7c,d for LB. For the *S. mutans* and *P. gingivalis* groups, ZOI was observed at all time points (1 h, 24 h, 48 h and 144 h) for the CHX-CaCl_2_ coated groups, whereas no ZOI was observed for the commercial mouthwash groups (Grps. 4 and 7; Figure 7). The ZOI for single and double CHX-CaCl_2_ coated Ti discs and LB were significantly different (*p* < 0.05) within groups for both pathogens tested (Figure 7).

## 3. Discussion

The proximity of arch bars and Leonard Buttons to the surgical incision site, coupled with biofilm, causing infection requires the need for strategic reduction in plaque biofilm volume. This could be achieved by utilising the novel CHX-CaCl_2_ coating technique described.

The Leonard Button and Ti discs’ elemental compositions were consistent with the literature [19,20] for metals used in maxillofacial surgery (>99% Titanium, Table 1). To standardise our novel CHX-CaCl_2_ coating procedure, the Ti discs and Leonard Buttons were coated with ethanol to enhance surface reactivity [21]. Both Ti discs and the LBs were successfully coated using surface crystallisation of CHX-CaCl_2_ crystals. Upon synthesis of the CHX-CaCl_2_ crystals, a rapid coprecipitation reaction occurred, forming a white precipitate with remarkable substantivity on Ti discs and LB (Figure 5). This reaction was due to the coordinating ability of the bisguanide group with calcium ions and with chlorine ions associated with the rate of formation [17]. Previous synthesis of CHXD with strontium chloride or zinc chloride demonstrated the aforementioned coordination ability, yielding antimicrobial crystals of dendritic nature with reduced cytotoxicity [22].

The CHX-CaCl_2_ crystals formed copiously on the Ti disc and Leonard Button surfaces, with various crystal sizes and evidence of particle coalescence (Figure 2b,c). An Ostwald ripening process may be responsible as larger CHX-CaCl_2_ crystals appear to grow at the expense of the smaller crystals through reducing the total free energy or by defect-related phenomena [23,24]. Luo et al. [13] reacted CHXD with CaCl_2_, together with differing amounts of gold nanorods, where crystal size and crystal number were correlated with gold nanorod addition, indicating a surface-crystallisation mechanism. The lapping of Ti discs in this study altered the surface-flaw size (Table 2), reducing visible surface crystallisation (Figure 2g–i) and drug release from 6 to 3 days (Figure 5). Fewer sites were, therefore, available for nucleation and crystal growth after lapping, supporting a surface crystallisation mechanism. Fine crystallites (5 µm) were growing in association with flaws on the lapped Ti surface (Figure 2g–i), which may also not have contributed to a long-lasting drug-release effect. Page and Sear [25] indicated the importance of grooving on the surface to control crystal growth when evaluating using computer simulations. The differing geometries and surfaces of wires and Leonard Buttons will, therefore, affect CHX-CaCl_2_ crystallisation and the effective wetting of the substrates. The rapid surface crystallization (<3 s) at the metal interface suggests good wetting, as it appeared crystallites grew by the extension of dendrites on the titanium surface (Figure 3d–f), ensuring rapid growth at the MMF site for effective and sustained drug release (Figure 5b). The importance of hydrophilicity of a surface is significant within the interaction between a solid state and the liquid state. Suematsu et al. [26] suggested that the rapid growth rate of dendritic NaCl crystals was due to growing on hydrophilic rather than hydrophobic surfaces.

Double coating of the Ti discs and Leonard Buttons using the novel CHX-CaCl_2_ mouthwash resulted in a doubling of the sustained CHXD drug release (Figure 5). There were changes to some of the crystal morphology for the layered crystallites, which had a partially formed wheat-sheaf appearance. This illustrated a more primary crystal growth from a central nucleus (Figure 2f). This may be due to the presence of residual reactants and their effects on the stoichiometry of secondary reactants. In particular, this simple multiple application of the novel rinse affords a double crystal-layer application on the Leonard Buttons, which is useful for maintenance or oral hygiene. Currently, a CHX digluconate mouthwash (0.2% CHX) is recommended as a postoperative mouthwash for craniomaxillofacial patients, which lacks substantivity and was antibacterially ineffective in the ZOI experiments and below minimum inhibitory concentration (MIC) after 20 min (Figure 7). The ZOI studies revealed that at time points associated with wound healing [27], the CHX-CaCl_2_ coated groups were effective against both aerobic and anaerobic pathogens (Figure 7a–d). This may significantly contribute to bacterial inhibition at orthopedic bars, plates and MMF sites. The addition of CHX-CaCl_2_ coatings on surgical fixatures and teeth following surgical procedures may, therefore, help in reducing infections and maintaining oral hygiene.

## 4. Materials and Methods

### 4.1. Energy Dispersive X-ray Spectroscopy

A titanium (Ti) sheet (1.2 mm depth, T60, Grade 4 Ti, Titanium services, Vourles, France) was laser cut (Lasercut Works Ltd., London, UK) into discs (8 mm diameter) to mimic the size of a Leonard Button (LB). The elemental composition of the Ti disc and Leonard Button (Lot 260785572, Synergy Health, Stryker, UK) was analysed using Energy Dispersive X-ray Spectroscopy (EDS) (Oxford Instruments, High Wycombe, UK) in a Scanning Electron Microscope (SEM) at an accelerating voltage of 30 kV, spot size 3.0, and with a 10 mm working distance. INCA ver.4.09 software (Oxford Instruments, Abingdon, UK) was used for EDS analysis.

### 4.2. Surface Coating of CHX-CaCl_2_ Particles

Twelve Ti discs were soaked in ethanol for 5 min in an ultrasonic bath and then dried in a 37 °C incubator (Benchmark Scientific Inc., Sayreville, NJ, USA) for 20 min. The Ti discs (Grp. 1, *n* = 3) were placed in a custom petri dish and coated using 1 mL (15 mg/mL) of CHXD (C6143, Lot WXBC6938V, Sigma-Aldrich, Dorset, UK) and left for 1 min. The amount of 1 mL of CaCl_2_ (0.33 M, C8106, Lot SLCD1523, Sigma-Aldrich) was then added using a Pasteur pipette and left for 1 min (23 °C), allowing the precipitation of CHX-CaCl_2_ crystals. Grp. 2 Ti discs (*n* = 3) were double coated by repeating this crystallisation process.

To assess the effect of the surface roughness, Ti discs (*n* = 3) were lapped manually using P240 silicon carbide paper (WS Flex 18C Hermes, Hamburg, Germany) for 2 min (Grp. 3). The lapped Ti discs were then single coated using the previous method. As a commercial comparison, (Grp. 4) Ti discs (*n* = 3) were coated with 2 mL of a commercial mouthwash (Corsodyl, 0.2%, CHX digluconate, Batch No 8480230, PL 44673/0059, Omega Pharma Manufacturing GmbH & Co, KG, Herrenberg, Germany) and left for 1 min according to the mouthwash protocol [28]. The CHX-CaCl_2_-coating process was repeated for Leonard Buttons, where Grp. 5 was single coated, Grp. 6 double coated and Grp. 7 was coated with 2 mL of the commercial mouthwash (Corsodyl) according to the previous protocols.

### 4.3. Scanning Electron Microscopy

The Ti discs and LB before and after CHX-CaCl_2_ coating were characterised using Scanning Electron Microscopy (SEM, FEI Inspect-F, Hillsboro, OR, USA). Ti discs and LB were gold-coated using a sputter coater (SC7620, Emitech, Chelmsford, UK) for 40 s at 20 mA, then viewed using SEM in the secondary electron imaging mode, with an accelerating voltage of 10 KV, spot size 3 and a working distance of 10 mm.

### 4.4. UV-Vis Spectroscopy

Coated Ti discs (Grp. 1–4) and LB (Grp. 5–7) specimens from Section 4.2. were transferred to universal tubes (Thermo Fisher Scientific, Swindon, UK, Lot M355760) containing 1 ml of artificial saliva (AS, pH 7) at 37 °C prepared according to Ten Cate et al. [29]. Ti discs and LB (Grp. 1–7) were relocated to tubes of fresh AS media at time intervals according to Table 3. Collected aliquots were analysed using UV-Visible Spectroscopy (UV-Vis) (Lambda 265, PerkinElmer, Waltham, MA, USA) to measure the chlorhexidine release at 254 nm. A calibration curve was obtained by measuring CHXD and CHX-Digluconate standards (0.5, 1, 3, 5, 10, 20, 30 and 50 ppm) and plotted as absorbance versus concentration.

### 4.5. X-ray Diffraction Analysis

For X-ray Diffraction (XRD) analysis, the precipitated CHX-CaCl_2_ particles (Section 4.2) were reacted in 2 mL Eppendorf tubes (*n* = 6), then centrifuged twice for 1 min at 90 rpm (Eppendorf AG, Hamburg, Germany). The supernatant was removed and replaced with deionised water and recentrifuged. The CHX-CaCl_2_ particles were then freeze-dried at −100 °C, at 0.009 mBar for 1 day (ScanVac CoolSafe Freeze Drying, Allerød, Denmark). The CHX-CaCl_2_ particles were analysed using an X’Pert Pro powder diffractometer (Panalytical B.V., Almelo, The Netherlands). The reflective mode used power at a 45 kV tension unit and 40 mA as a current unit at a 6° angle, and the data were collected with an X’Celerator detector from 5° to 70° 2 Theta, with a step size of 0.005°.

### 4.6. Non-Contact Light Profilometry

Surface analysis of Ti discs (Grps. 1 and 2) was analysed using a non-contact 3D light profilometer (Proscan 2000, Scantron, Taunton, UK), where an S16/3.5 Chromatic sensor (Stil S.A., Aix-en-Provence, France) scanned at 100 Hz frequency with a 3.5 mm measuring range and a 75 nm axial resolution. The digitisation, image analysis and the mean area surface roughness (Ra) were analysed using the dedicated software (Proscan 2000, ver.2.1.8.8+ software, Proform ver.1.41 software, Scantron Industrial Products Ltd., Taunton, UK). To ensure maximum sensitivity of light was achieved for calibration purposes, a dark background measurement was carried out prior to scanning. The Ti disc and lapped Ti disc’s scanned surface was limited to 4 × 4 mm, at step size of 5 µm. The surface mapping was retrieved from the scanned sample and mean x and y measurements automated via the Proform ver.1.41 software through 801-line counts in x and y.

### 4.7. Preparation of Pathogens

S. mutans (NCTC 10449) strains were cultured on Tryptone Soy Agar (TSA) (Lot 2426701, Oxoid, Basingstoke, UK) on agar plates placed in 37 °C aerobic incubation for 2 days. The 2-day colonies were inoculated in 10 mL Tryptone soya broth (TSB) (Oxoid, Basingstoke, UK) and placed for 24 h incubation. The *P. gingivalis* strain (W50) was cultured on Blood Agar (BA) base No. 2 (lot 2359559, CM0271 Oxoid, Basingstoke, UK) agar plates that were supplemented with 5% defibrinated horse blood in an anaerobic incubator (Don Whitley, UK) at 37 °C for 3 days. The 3–5 days colonies were inoculated in 10 mL Brain Heart Infusion (BHI). The broth was supplemented with equal-volume amounts of 100 µL of Vitamin K and hemin, then incubated overnight.

### 4.8. Kirby–Bauer Test (Zone of Inhibition)

The optical density of pathogens in Section 4.7 was achieved by diluting the culture broth to 0.1 in a Bio Photometer at 600 nm (Eppendorf AG, Hamburg, Germany). The bacterial suspension was then used for calculating Colony forming unit (Cfu), where broth inoculum was standardised at 6.36 × 106 Cfu per mL. 100 μL bacterial suspension was spread onto TSA plates for *S. mutans* strains; BA plates for *P. gingivalis*, and holes were made using a sterilised cork borer (0.9 cm diameter). The CHXD and CHX digluconate dilution standards of 100 μL were pipetted into the holes, then incubated overnight at 37 °C. The diameter (mm) of the zone was measured across the formed zone using a ruler at 24 h to measure the inhibition of bacterial growth. A standard curve of concentration (ppm) versus the distance of the zone was used to calculate the zone of inhibition. This process was repeated with samples obtained from the drug-release time points (1 h, 24 h, 48 h and 144 h) for coated Ti discs (Grps. 1, 2 and 4) and Leonard Buttons (Grps. 5–7), with AS used as a control. Significant differences within groups (*p* < 0.05) were measured using a one-way ANOVA (Tukey test; Sigma plot ver.12.5, Systat Software Inc., San Jose, CA, USA).

## 5. Conclusions

There is increasing evidence that periodontal disease is heavily implicated in the progression of surgical site infection in intraoral incisions used in osteosynthesis of the mandible. The proximity of the dental fixatures to the surgical incision and the dependent position of the incision could potentially promote colonization of the wound from bacteria previously coating the fixation material. This work illustrates for the first time the rapid and successful coating of Leonard Buttons and Ti discs using a CHX-CaCl_2_ mouth rinse and its responsive surface crystallisation. The ability to control and sustain CHXD release from this novel coating, and its antibacterial effectiveness, makes this an ideal adjunct following surgical procedures to maintain oral hygiene and prevent infections. The previous encouraging laboratory results will be further tested in a prospective trial to demonstrate the clinical efficacy of this new technology.

## Figures and Tables

**Figure 1 ijms-24-09801-f001:**
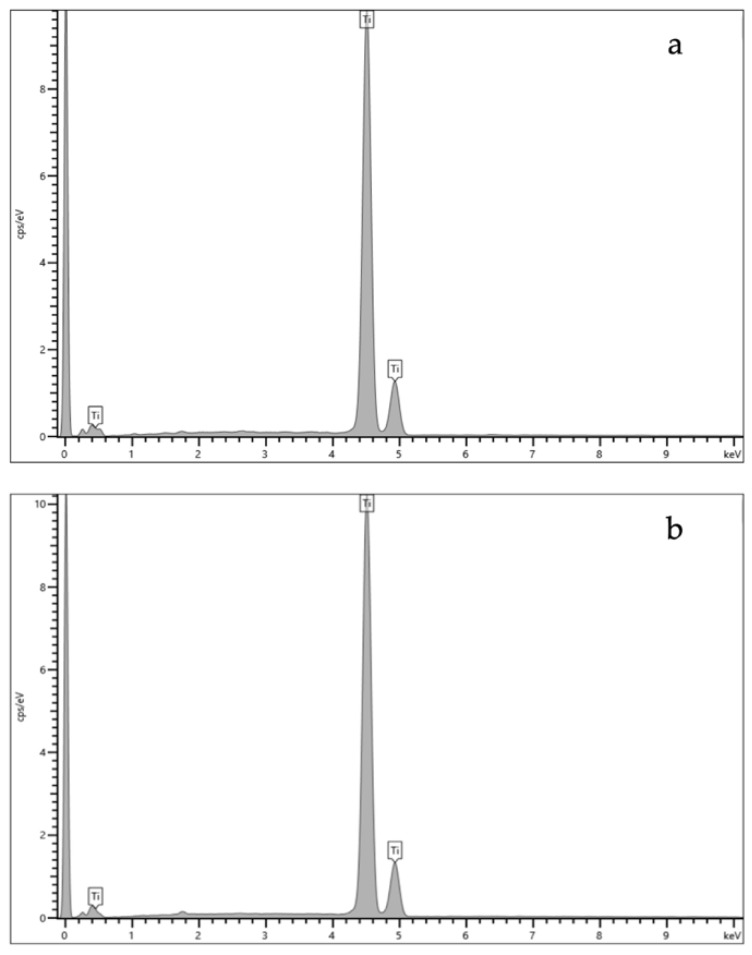
EDS spectra of (**a**) Ti disc; (**b**) Leonard Button.

**Figure 2 ijms-24-09801-f002:**
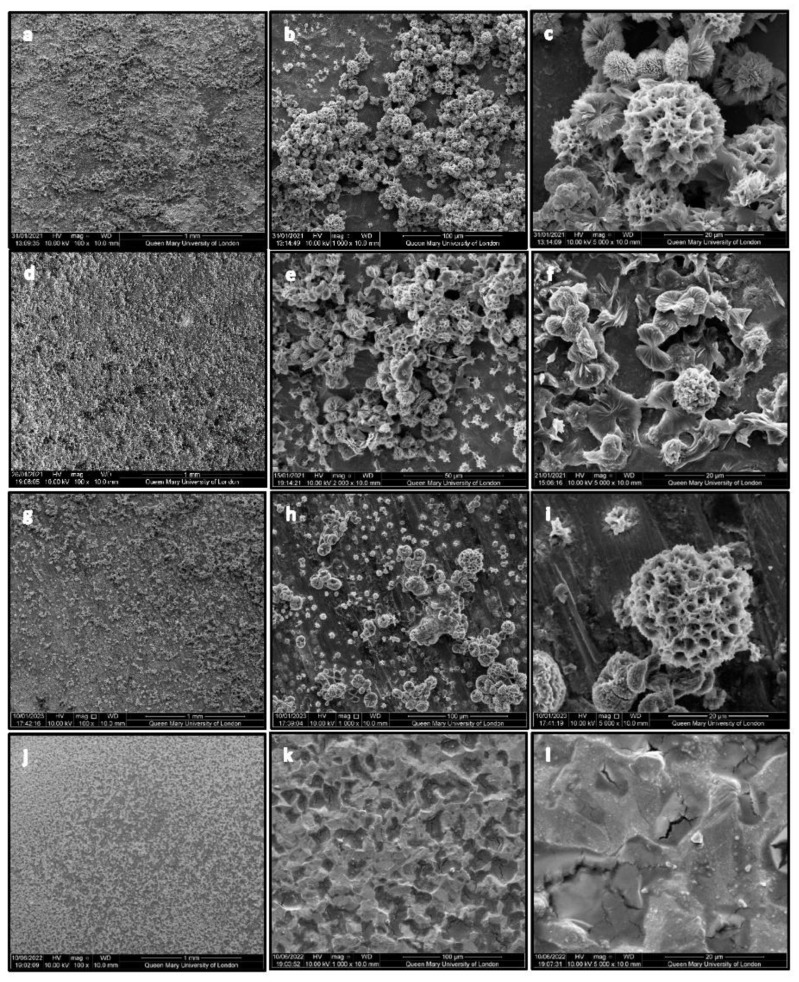
SEM photomicrographs of a copious coating of CHX-CaCl_2_ crystals on single coated Ti disc (**a**–**c**); double-coated Ti disc (**d**–**f**); lapped Ti disc (**g**–**i**) and Ti disc coated with Corsodyl (**j**–**l**).

**Figure 3 ijms-24-09801-f003:**
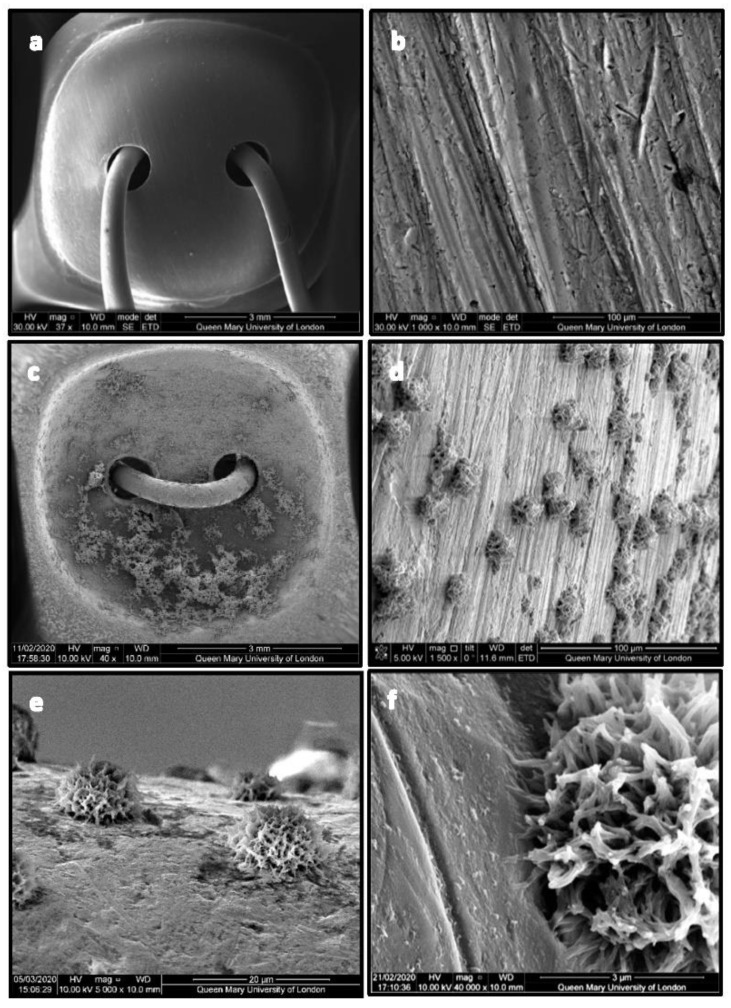
SEM photomicrographs of LB uncoated (**a**,**b**) and single coated with CHX-CaCl_2_ particles (**c**–**f**).

**Figure 4 ijms-24-09801-f004:**
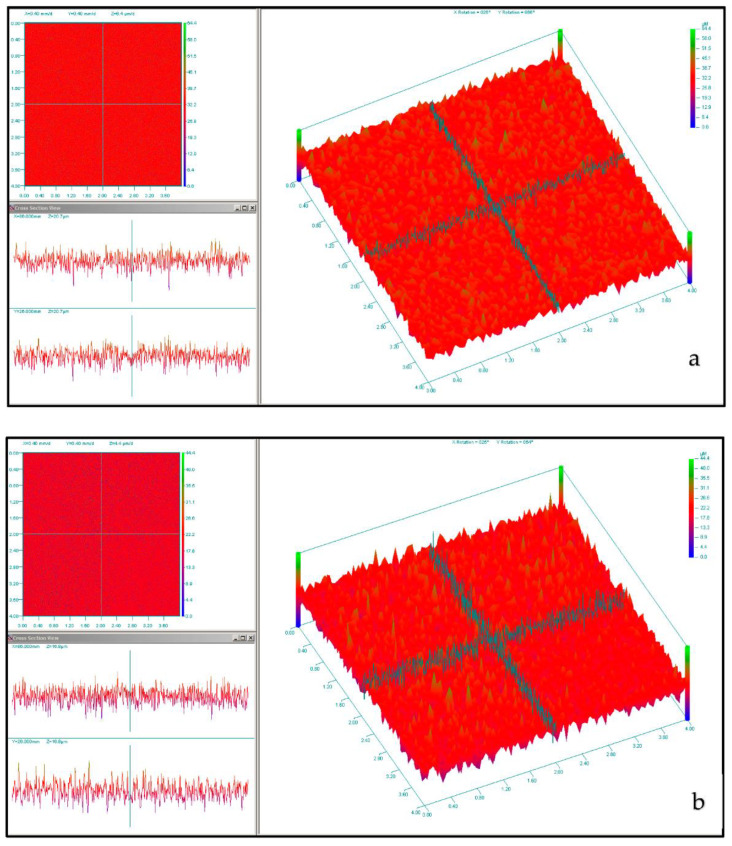
Light profilometry results displaying the grid and primary mapping of (**a**) Ti disc and (**b**) Lapped Ti disc.

**Figure 5 ijms-24-09801-f005:**
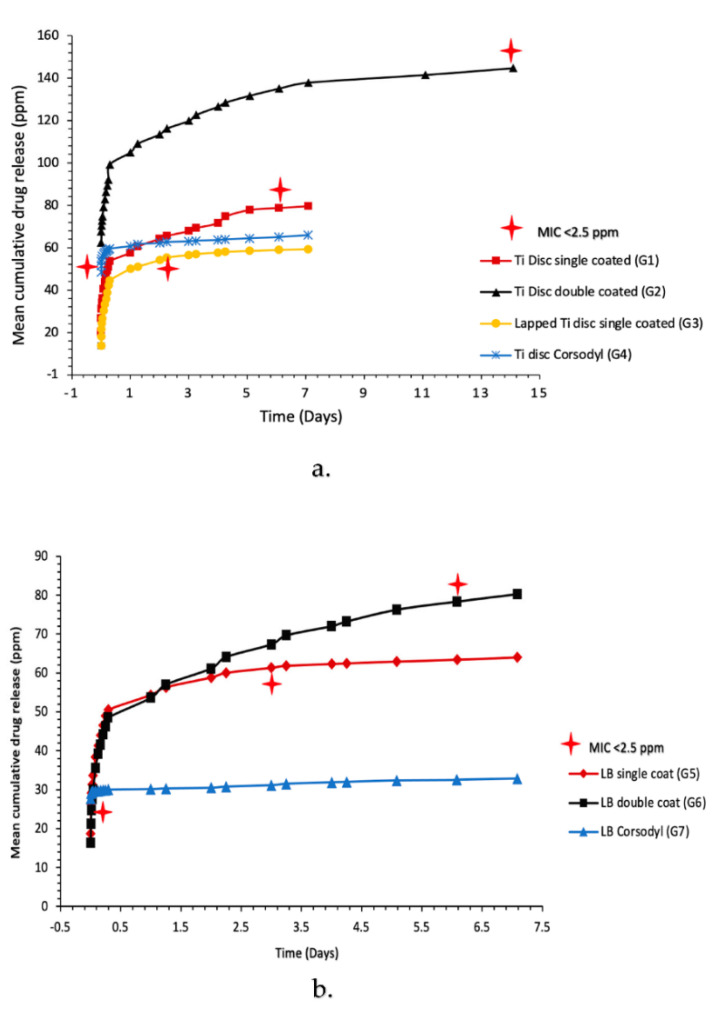
Mean cumulative CHXD drug release of: (**a**) of Ti disc (Grps. 1–4); (**b**) LB (Grps. 5–7). Plot values represent *n* = 3 readings at each time point.

**Figure 6 ijms-24-09801-f006:**
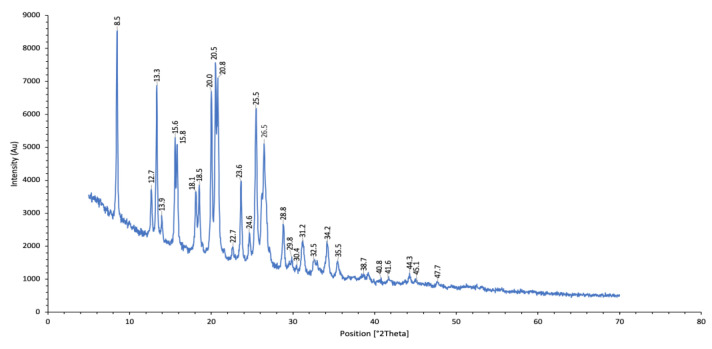
XRD plot of CHX-CaCl_2_.

**Figure 7 ijms-24-09801-f007:**
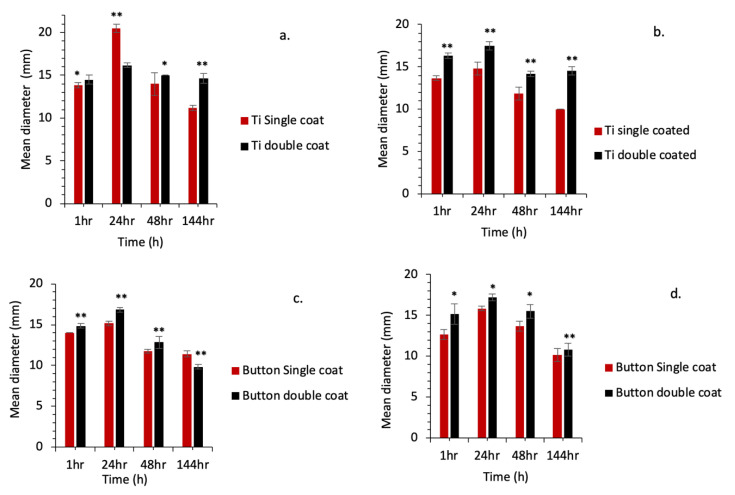
ZOI results for *P. gingivalis* strains on (**a**) coated Ti discs; (**c**) coated LB and for *S. mutans* strains on (**b**) coated Ti discs; (**d**) coated LB. * indicates a significant difference within groups (*p* < 0.05). ** indicates a highly significant difference within groups (*p* < 0.001).

**Table 1 ijms-24-09801-t001:** Composition of Titanium disc and Leonard Button (atomic %).

	Mean (SD) of Elements (Atomic %)				
Spectrum	Si	Ti	Cr	Fe	Ga
Titanium disc		99.57 (0.38)		0.13 (0.11)	0.31 (0.27)
Leonard Button	0.52 (0.20)	99.26 (0.34)	0.04 (0.08)	0.28 (0.11)	0.30 (0.08)

**Table 2 ijms-24-09801-t002:** Mean surface roughness (Ra) of specimens from Groups 1 and 3.

Samples	Mean X (µm)	Mean Y (µm)	Average (µm)
Ti disc (Group 1)	4.2	4.2	4.2
Lapped Ti disc (Group 3)	3.3	3.3	3.3

**Table 3 ijms-24-09801-t003:** UV-Vis schedule for the release assay.

Week	Day	Time Interval (min/h)	
1st Week	1st Day (Monday)	5 min	~10:05 (10:00 am start)
		10 min	~10:10
		20 min	~10:20
		40 min	~10:40
		60 min (1 h)	~11:00
		120 min (2 h)	~12:00
		180 min (3 h)	~13:00
		240 min (4 h)	~14:00
		300 min (5 h)	~15:00
		360 min (6 h)	~16:00
		420 min (7 h)	~17:00
	2nd Day (Tuesday)	2 readings	~10:00 and ~16:00
	3rd Day (Wednesday)	2 readings	~10:00 and ~16:00
	4th Day (Thursday)	2 readings	~10:00 and ~16:00
	5th Day (Friday)	2 readings	~10:00 and ~16:00
	6th Day (Saturday)	1 reading	~12:00
	7th Day (Sunday)	1 reading	~12:00
2nd Week	Monday	1 reading	~12:00
	Friday	1 reading	~12:00

## Data Availability

The data presented in this study are available on request from the corresponding author.

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
