# Peer review of "Synthesis of Novel Antimicrobial CHX-CaCl_2_ Coatings on Maxillofacial Fixatures for Infection Prevention"

_ijms, 2023, doi:10.3390/ijms24129801_

Round 1

Reviewer 1 Report

I would like to recommend this manuscript for publication after minor revision:

1. The words in Figure 1 are too small, I cant see them clearly. And the figures appear to have traces of irregular dragging causing text deformation. Please provide the original image

2. The data in Figure 5 is single data or average data, it should be presented as mean±SD, N=?

3. Please mark what each peak mean in Figure 6.

4. Please add the Bacterial experiments in the manuscript.

5. Here are some antibacterial and antiviral studies recommended for Introduction in the paper:

(1) Adv. Fiber Mater., 2022, 4, 76

(2) Adv. Fiber Mater., 2022, 4, 119

(3) Adv. Fiber Mater., 2022, 4, 89

(4) Adv. Fiber Mater., 2022, 4, 1304

Reviewer 2 Report

This study synthesized an antimicrobial coating of CHX-CaCl2 particles on the maxillomandibular fixation to improve oral hygiene and reduce maxillofacial infection rates.

The manuscript is clear, relevant, and presented in a well-structured manner. Furthermore, it is scientifically sound and presents an appropriate experimental design to test the hypothesis. Nearly 50% of the cited references are from the last 5 years. The results are reproducible depending on the details given in the methods. Figures, tables, and images are appropriate, display the data correctly, and are easy to interpret and understand. Data are interpreted appropriately and consistently throughout the manuscript. A correct statistical analysis is carried out and the conclusions are consistent with the evidence and the arguments presented.

Reviewer 3 Report

"Synthesis of Novel Antimicrobial CHX-CaCl2 Coatings on Max- 2 illofacial Fixatures for Infection Prevention." is an article about the efficacy of gold nanorod coated with Chlorhexidine cristal for prevenction of bacterial infections in patients upon maxillofacial surgery. The article is well written and clear to understand (except for the big numbers of acronyms that were not explained so it was a bit tricky to understand everything before reaching the methodology section).

I recommend the article for publication on IJMS.

Following my considerations: 

Acronyms should be explained the first time they're found in the text. For example EDS is never described (line 68/69) as well as SEM and so on... or CHX even if in this last case it can be deduced from the text. The full meaning of acronyms will help even the less expert to full understand the text. 

Please check the table 2 since it is splitted between pages 4 and 5. Guess it would be better if it is in the same page (for a better visualization).

Pls check the bibliography: some articles have the first letter of every word in Capital (reference 10)  while other not and some hyperlink are active while other not (check the doi).
